# Future Gradient Descent for Adapting the Temporal Shifting Data Distribution in Online Recommendation Systems

**Mao Ye**[1]  **Ruichen Jiang**[1]  **Haoxiang Wang**[2]  **Dhruv Choudhary**[3]  **Xiaocong Du**[3]  **Bhargav Bhushanam**[3]

**Aryan Mokhtari**[1]  **Arun Kejariwal**[3]  **Qiang Liu**[1]

[1]The University of Texas at Austin.
[2]The University of Illinois at Urbana-Champaign.
[3]Meta.

## Abstract

One of the key challenges of learning an online recommendation model is the temporal domain shift, which causes the mismatch between the training and testing data distribution and hence domain generalization error. To overcome, we propose to learn a meta future gradient generator that forecasts the gradient information of the future data distribution for training so that the recommendation model can be trained as if we were able to look ahead at the future of its deployment. Compared with Batch Update, a widely used paradigm, our theory suggests that the proposed algorithm achieves smaller temporal domain generalization error measured by a gradient variation term in a local regret. We demonstrate the empirical advantage by comparing with various representative baselines.

## 1 INTRODUCTION

The web-scale recommendation system is one of the most important modern machine learning applications that provides personalized content to billions of users from inventories of billions of items. These recommendation models have been rapidly growing in both computation and memory in the past few years due to wider-deeper networks and the use of sparse embedding layers. [He et al., 2020, Peng et al., 2021, Ye et al., 2020, Zhang et al., 2020] have demonstrated the importance of updating the recommendation periodically (e.g., in a daily/weekly basis) as new data arrives to avoid the model being stale in a domain shifting environment.

Designing such a periodical updating pipeline is non-trivial: the algorithm needs to achieve a good balance of consolidating the long-term memory (ensuring the useful past knowledge is preserved) and capturing short-term tendency, which is valuable for near future prediction [Deng et al., 2021, Peng et al., 2021, Zhang et al., 2020]. Algorithms

can be categorized into two groups: 1. The *sample-based* approaches maintain a reservoir to reuse observed historical examples to preserve long-term memory [Diaz-Aviles et al., 2012]. Several heuristics are developed to select past examples via balancing the prioritizing of recency and forgetting [Chen et al., 2013, Qiu et al., 2020, Wang et al., 2018, Zhao et al., 2021]; 2. The *model-based* approaches maintain the long-term memory by transferring knowledge between the past and the current model checkpoints via knowledge distillation [Mi et al., 2020, Wang et al., 2020, Xu et al., 2020] and model fusion [Peng et al., 2021, Zhang et al., 2020].

In this paper, we provide a novel perspective by framing the problem of learning under shifting domains as a *temporal domain generalization problem*. We observe that the crux lies in the mismatch between the distribution of the training examples and the distribution of the testing example on which the model is deployed for recommendations. From this perspective, existing approaches mitigate such the crux by the distribution mismatch in an *indirect* way by training a robust model that is less vulnerable to the shift of domain in the near future by making it a master at both short-term and long-term signals. More precisely, we propose a more *direct* solution towards the temporal domain generalization problem based on forecasting the future information for training. Consider the ideal case that we are able to access the data distribution in the near future when the model is deployed, simply training the model by gradient descent using examples drawn from the future data distribution should be desirable (see 'Ideal Update' in Fig 2). In the real world when such future information is unavailable, we propose to train a meta future gradient generator to forecast the gradient of the future examples so that the recommendation model is trained as if we were able to look ahead at the future (i.e., 'FGD Update' in Fig 2). In addition to the sample-based and model-based approach, our method is *optimizer-based* in that the trainer of the recommendation model is improved.

In theory, we frame the problem as an online learning problem in which the temporal domain generalization error is captured by the gradient variation term [Chiang et al., 2012,

*Accepted for the 38th Conference on Uncertainty in Artificial Intelligence* (UAI 2022).

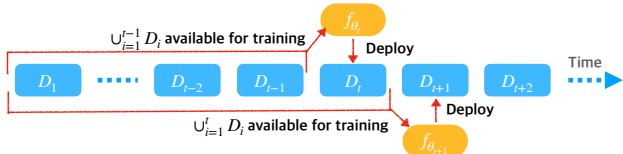

Figure 1: Illustration of the temporal domain generalization problem where the distribution of the training set of $f_{\theta_t}$ mismatches the distribution of its test set $D_t$ at time $t$.

Rakhlin and Sridharan, 2013] in a local regret [Hazan et al., 2017]. We provide a theoretical understanding of why the proposed algorithm improves over batch-update, a widely used training pipeline [Wang et al., 2020] and show that our method is able to achieve a similar regret as that of a fixed meta future gradient generator oracle. Empirically, we compare our approach against several representative sample/model-based approaches and observe considerable performance improvement.

**Notation.** We denote the integer set $\{1, 2..., b\}$ by $[b]$. Moreover, $\| \cdot \|$ denotes the $\ell_2$ vector norm, $\| \cdot \|_1$ denotes the $\ell_1$ vector norm, and $S_b = \{a \in \mathbb{R}^b : a_i \geq 0, \|a\|_1 = 1\}$ is the probability simplex set.

## 2 PROBLEM AND BACKGROUND

**Temporal Domain Generalization.** Consider an online classification problem with the feature space $\mathcal{X}$ and the label space $\mathcal{Y}$. Our goal is to learn an accurate prediction model $f_\theta : \mathcal{X} \to \mathcal{Y}$ parameterized by $\theta \in \Theta$ from a stream of datasets $D_1, \ldots, D_T$ in $T$ consecutive rounds. Specifically, at round $t$, we choose a model parameter $\theta_t \in \Theta$ and deploy our prediction model $f_{\theta_t}$. Then we observe the dataset $D_t$ with $n_t$ labeled examples $D_t = \{(x_t^{(i)}, y_t^{(i)})\}_{i=1}^{n_t}$ drawn from certain data distribution $\mathcal{P}_t$, where $x_t^{(i)}$ are the input features and $y_t^{(i)}$ is the associated label. Thus, for a given loss function $\ell : \mathcal{X} \times \mathcal{Y} \to \mathbb{R}_+$, the empirical loss of our prediction model at time $t$ is given by $r_t(\theta) = \mathbb{E}_{(x,y) \sim D_t} \ell(f_\theta(x), y)$. Moreover, we consider the situation where the data distribution $\mathcal{P}_t$ (i.e., domain) is gradually changing over time. A natural performance metric for our learning algorithm is the temporal average of the test loss suffered by the prediction model:

$$\frac{1}{T} \sum_{t=1}^{T} r_t(\theta_t). \quad (1)$$

We remark that $T$, which denotes the total number of rounds in the online process, is typically large in practice.

The key challenge is the temporal domain generalization. Indeed, at time $t$ we train our prediction model $f_{\theta_t}$ using the observed examples $\cup_{i \in \{0,1,\ldots,t-1\}} D_i$ and due to temporal shift of domain, the distribution of the test set does not match the distribution of its training set. Such mismatch of

**Algorithm 1** Batch and Incremental Update

**Input:** The learning rate $\eta$ for updating the parameter $\theta$.
**for** $t \in [T]$ **do**
    Deploy the prediction model $f_{\theta_t}$ with parameter $\theta_t$.
    Collect the new dataset $D_t$.
    Initialize $\theta_{t+1}$.
    **while** $\|\frac{1}{b} \sum_{i=0}^{b-1} \nabla r_{t-i}(\theta_{t+1})\| \geq \delta$ **do**
        $\theta_{t+1} \leftarrow \theta_{t+1} - \eta \frac{1}{b} \sum_{i=0}^{b-1} \nabla r_{t-i}(\theta_{t+1})$.
    **end while**
**end for**

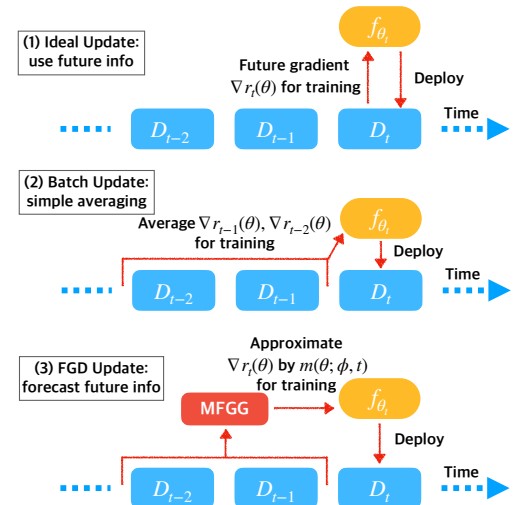

Figure 2: Comparing (1) the ideal update where the future information $\nabla r_t$ can be accessed at training time; (2) the batch update; and (3) our proposed approach.

the training and testing domains results in domain generalization error. See Fig 1 for an illustration.

Our formulation is motivated by the online recommendation systems that aim to advertise items to users given user features. The domain is gradually changing because of the flux in the content that gets continuously added/removed from the system [He et al., 2014, Ye et al., 2020]. As the recommendation model needs to be deployed for serving, it is hard to update its parameter in real time [Cervantes et al., 2018, Peng et al., 2021, Wang et al., 2020]. The training process is thus discretized in which the model parameter is updated periodically with the hope that it generalizes well in its test domain.

**Batch and Incremental Update.** Batch Update (BU) [Hazan et al., 2017, Wang et al., 2020] is a widely used updating pipeline for training the recommendation model in temporally shifting domains. At each time $t$, the model parameters are updated using the gradient of the averaged losses $r_t, \ldots, r_{t-b+1}$, where $b$ is a time window size indicating how many observed data are used. BU with $b = 1$ is also named as Incremental Updating (IU). We summarize the pipeline in Algorithm 1, where $\nabla r_s$ with $s \leq 0$ is defined

---

**Algorithm 2** Future Gradient Descent

---

**Input:** The learning rate $\eta, \eta_\phi$ for updating the model parameter $\theta$ and $\phi$. The initial trajectory buffer $B$.
**for** $t \in [T]$ **do**
    Deploy the prediction model $f_{\theta_t}$ with parameter $\theta_t$. Then collect the new dataset $D_t$.
    Initialize the parameter of MFGG $\phi_{t+1}$.                               $\triangleright$ Initialization of $\phi_{t+1}$ is user-specific.
    **for** Inner loop iteration $k \in K$ **do**                                  $\triangleright$ Update the meta network.
        $\phi_{t+1} \leftarrow \phi_{t+1} - \eta_\phi \sum_{\theta \in B} \nabla_\phi \|m(\theta; \phi_{t+1}, t) - \nabla r_t(\theta)\|^2$.       $\triangleright$ May replace with the mini-batch version.
    **end for**
    Initialize the trajectory buffer $B = \emptyset$ and model parameter $\theta_{t+1}$. $\triangleright$ Initialization scheme of $\theta_{t+1}$ is specified by user.
    **while** $\|m(\theta_{t+1}; \phi_{t+1}, t+1)\| \geq \delta$ **do** $\triangleright$ Alternatively, we may run gradient descent with a fixed number of iterations.
        $\theta_{t+1} \leftarrow \theta_{t+1} - \eta m(\theta_{t+1}; \phi_{t+1}, t+1)$.                     $\triangleright$ May replace with the mini-batch version.
        $B \leftarrow B \cup \{\theta_{t+1}\}$                    $\triangleright$ Alternatively, we may update the trajectory buffer $B$ every a few iterations.
    **end while**
**end for**

---

as 0. Also see an illustration of BU with $b = 2$ in the second plot of Fig. 2. It is noteworthy that the initialization scheme of $\theta'_{t+1}$ for the updating at each time is problem-dependent and user-specified. For example, we can set $\theta'_{t+1} = \theta_t$ of $\theta'_{t+1} = \theta_{t-b+1}$ if we consider one-pass training setting [Du et al., 2021, Ye et al., 2020, Zheng et al., 2020].

## 3   METHOD

Recall that our goal is to learn $f_{\theta_t}$ that gives accurate prediction on $D_t$ (i.e. achieves small $r_t$). Think of the ideal world where we were able to access the data $D_t$ at the near future during the training time of $f_{\theta_t}$, a simple while promising approach is to apply gradient descent using $\nabla r_t$ (see the first plot in Fig. 2). In the real case where the future information $\nabla r_t$ is no more available, we propose to learn a meta future gradient generator (MFGG) that forecasts $\nabla r_t$ given the observed data $\cup_{i=1}^{t-1} D_i$; see the third plot in Fig 2.

**Architecture of MFGG.** MFGG models $\nabla r_t(\theta)$ as an non-linear functional auto-regressive time series model [Bosq, 2000]. It approximates $\nabla r_t(\theta)$ by aggregating the gradient based on the latest $b$ losses $\sum_{i=0}^{b-1} a_i(D_{t-b}, ..., D_{t-1}) \nabla r_{t-1-i}(\theta)$ where the coefficient of the linear combination $a_i(D_{t-b}, ..., D_{t-1})$ is a neural network given by the following computation graph.

$$e_{i,j} = \text{Embd}(x_j^{(i)}) \in \mathbb{R}^{d_1}$$
$$e_j = \sum_{i \in [n_j]} e_{i,j} \in \mathbb{R}^{d_1}$$
$$z = \text{Self Attention}(e_{t-b}, ..., e_{t-1}) \in \mathbb{R}^{d_2 \times b}$$
$$a = \text{Softmax} \circ \text{MLP}(z_{t-b}, ..., z_{t-1}) \in \mathbb{R}^b$$

Here Embd denotes the embedding layer that maps the categorical feature into a continuous embedding space (the continuous feature remains the same in this layer); Self Attention denotes the self attention layer [Vaswani et al., 2017]; MLP denotes the multi-layer perception. MFGG first extracts the domain features $e_j$ over $j \in \{t - b, ..., t - 1\}$

of the last $b$ domains) and the self attention then encodes the interaction between the domain features, of which the outcomes are fed into the subsequent layers to calculate the coefficient $a$. The softmax layer is option and regularizes $a$ to be in a probability simplex $S_b$ and hence ensures the magnitude of the generated gradient is within a proper range. Suppose $\phi$ unions all the parameters, we denote MFGG as $m(\theta; \phi, t)$. In practice, we can simply replace $D_j$ with its mini-batch samples $\hat{D}_j$, which gives a stochastic gradient version for updating.

**Optimization of MFGG.** We use the squared $\ell_2$ loss $\|m(\theta; \phi, t) - \nabla r_t(\theta)\|^2$ for measuring the prediction error of $m(\theta; \phi, t)$ at time $t$. Such error depends on both $\phi$, the parameter of MFGG and $\theta$, the parameter of recommendation model used for calculating the gradient. We are more interested in make MFGG accurate at a small subset of the model parameter space $\Theta$ in which $\theta$ gives a recommendation model with good performance. We thus only apply the $\ell_2$ loss on the (sub-sampled) optimization trajectory of $\theta$, which we denoted as $B$. That is, we learn $m(\theta; \phi, t)$ by apply gradient descent on

$$\sum_{\theta \in B} \|m(\theta; \phi, t) - \nabla r_t(\theta)\|^2.$$

Note that here when calculating the gradient of $\phi$, $\theta$ is viewed as a constant and hence the differentiation of $\phi$ at $\theta$ does not applied. Algorithm 2 summarizes the detailed procedure. Again, a mini-batch version of $m(\theta; \phi, t)$ and $\nabla r_t(\theta)$ can be used during the training of MFGG. In practice at $t \leq b$, we don not have enough historical data to compute MFGG, we can simply use IU for training (alternatively, data for offline training can be used instead). Since our approach uses the MFGG to predict the gradient of the loss on the unobserved future data, we name it Future Gradient Descent (FGD).

**Extension to a smoothed loss.** In practice, one might be interested in a smoothed version of performance metric as it

is observed to be a potentially more robust evaluation metric in practice [He et al., 2014]. More precisely, consider the loss function

$$\frac{1}{T}\sum_{t=1}^{T}\left[\frac{1}{w}\sum_{i=0}^{w-1}r_{t-i}(\theta_t)\right], \qquad (2)$$

where $r_s$ is identically zero for $s \le 0$. This smoothed loss in (2) uses a sliding window with width $w$ over the previous datasets $\cup_{i=0}^{w-1}D_{t-i}$ when evaluating. We are mainly interested in the standard metric (1) but when (2) is considered, we can simply generalize FGD by replacing $m(\theta; \phi, t)$ by

$$\bar{m}(\theta; \phi, t) = \frac{1}{w}\left(m(\theta; \phi, t) + \sum_{i=1}^{w-1}\nabla r_{t-i}(\theta)\right),$$

when training $\theta$. Here $\nabla r_s$, $s \le 0$ is defined 0. We refer readers to Algorithm **??** in Appendix **??** for the details. In the rest of the paper, we focus on the smoothed version of loss as it is more general.

Before moving forward, we emphasize the difference between the two window sizes $b$ and $w$ that appear in the BU/FGD and in the definition of (2), respectively. In some sense, $b$ corresponds to the number of recently observed datasets used for training the model. While, $w$ represents the number of datasets used for testing the model.

# 4 THEORY

In this section, we study the advantage of the proposed FGD over BU and IU theoretically using recent advances in non-convex online learning. Specifically, we show that FGD is able to perform better than BU and IU in terms of the so-called *local regret* [Hallak et al., 2021, Hazan et al., 2017], which measures the algorithm's performance by comparing it with the best one can achieve in hindsight.

## 4.1 LOCAL REGRET

To upper bound the average loss in (1) in a changing environment, one standard approach is to study the average dynamic regret [Zinkevich, 2003]:

$$\frac{1}{T}\sum_{t=1}^{T}[r_t(\theta_t) - \min_{\theta \in \Theta} r_t(\theta)], \qquad (3)$$

which uses the global minimum of $r_t$ as a benchmark when evaluating the performance at time $t$. However, in modern recommendation systems the prediction model $f_\theta$ is given by a deep neural network, and thus the resulting loss function $r_t(\theta)$ is highly non-convex. This means finding an approximate global minimum of $r_t$ is computationally intractable, making it hopeless to derive any meaningful bound on the average dynamic regret in (3). To remedy this issue, we adopt the notion of *local regret* proposed by Hazan et al. [2017]. Specifically, given $\{\theta_t\}_{t=1}^T$ generated by an

online learning algorithm, the average *local regret* is defined as

$$\mathfrak{R}(T) := \frac{1}{T}\sum_{t=1}^{T}\|\nabla r_t(\theta_t)\|^2. \qquad (4)$$

Compared with (3), in (4) we evaluate the model parameters in terms of the first-order stationarity, and thus it can be viewed as the non-convex counterpart of the dynamic regret in (3). In particular, a small value of $\mathfrak{R}(T)$ implies a small gradient on average, suggesting that the algorithm achieves near-optimal performance locally in the long run.

More generally, when the smoothed loss (2) is considered, one can use the average $w$-*local regret* accordingly as in [Hazan et al., 2017]:

$$\mathfrak{R}_w(T) := \frac{1}{T}\sum_{t=1}^{T}\|\nabla u_{w,t}(\theta_t)\|^2,$$

where we evaluate $\theta_t$ using the smoothed loss function $u_{w,t}(\theta) := \frac{1}{w}\sum_{i=0}^{w-1}r_{t-i}(\theta)$. In the following, we will focus our analysis on $\mathfrak{R}_w(T)$, as choosing $w = 1$ also covers the standard local regret in (4).

## 4.2 REGRET OF BATCH UPDATE

In [Hazan et al., 2017], the authors analyzed the average $w$-local regret $\mathfrak{R}_w(T)$ for BU. We recall their result below but offer a different interpretation from the domain generalization perspective.

**Proposition 1** ([Hallak et al., 2021, Hazan et al., 2017]). *With the choice of the window size $b = w$, the $w$-local regret incurred by BU in Algorithm 1 satisfies*

$$\mathfrak{R}_w(T) \le \underbrace{2\sum_{t=1}^{T}\|\nabla u_{w,t-1}(\theta_t)\|^2/T}_{\text{optimization error}} + \underbrace{2V_w(T)/w^2}_{\text{domain generalization}}$$

$$\le 2\delta^2 + \frac{2}{w^2}V_w(T),$$

*where $V_w(T) = \frac{1}{T}\sum_{t=1}^{T}\sup_{\theta}\|\nabla r_t(\theta) - \nabla r_{t-w}(\theta)\|^2$.*

*Furthermore, if $\|\nabla r_t(\theta)\| \le M < \infty$ for all $\theta \in \Theta$ and $t \ge 0$, choosing $\delta = O(1/w)$ gives $\mathfrak{R}_w(T) = O(1/w^2)$, which is minimax optimal.*

The previous works [Hallak et al., 2021, Hazan et al., 2017] are interested in the worst-case guarantee of the BU algorithm, and the result in Proposition 1 only serves as an intermediate result. However, we observe that this regret bound also offers interesting insights from the perspective of domain generalization. To be specific, we can decompose it into two terms:

*The optimization error*: this is due to the fact that we only seek a $\delta$-approximate stationary point of the smoothed training loss function $u_{w,t-1}(\theta)$ at round $t$. It is controllable

in the sense that $\delta$ can be made arbitrarily small by running more iterations of gradient descent. Indeed, under standard smoothness assumption on $r_i$, we can achieve $\|\nabla u_{w,t-1}(\theta_t)\| \leq \delta$ within $O(\delta^{-1})$ iterations. The optimization error term thus corresponds to how well we train the recommendation model in each round.

*The domain generalization error*: this is due to the fact that the the test set $\cup_{i=0}^{w-1} D_{t-i}$ for evaluating $\theta_t$ is different from the training set $\cup_{i=1}^{w} D_{t-i}$. It is typically the dominant term in the regret bound and will not vanish even when $\delta = 0$. In some sense, it captures the level of variability in the data distributions, similar to the gradient variation term in [Chiang et al., 2012, Rakhlin and Sridharan, 2013]. We also note that the domain generalization error decreases w.r.t. $w$. This is because when $w$ increases, the overlap between the training set and the test set becomes larger (i.e., the training set and the test set deviate less)[1].

In summary, the optimization error term characterizes how well our model performs on the training set, while the domain generalization error term characterizes how much the test set deviates from the training set.

**Comparison with other measure of domain divergence.** In Proposition 1, the domain discrepancy is characterized in terms of the gradient variation (i.e, how much the gradient of the loss functions differs). Some other domain discrepancy measures have also been proposed. Examples include the $\mathcal{H}$-divergence [Kifer et al., 2004] between $D$ and $D'$ defined as $d_{\mathcal{H}}(D, D') = \sup_\theta \|\mathbb{E}_\mathcal{D} \ell(f_\theta(x), y) - \mathbb{E}_{\mathcal{D}'} \ell(f_\theta(x), y)\|$ and the $\mathcal{H}\Delta\mathcal{H}$ divergence [Ben-David et al., 2010] defined as $d_{\mathcal{H}\Delta\mathcal{H}}(D, D') = \sup_{\theta,\theta'} \|\mathbb{E}_D \ell(f_\theta, f_{\theta'}) - \mathbb{E}_{D'} \ell(f_\theta, f_{\theta'})\|$. Overall, the commonly used divergence measures share the general form of $\sup_\theta \|\mathbb{E}_\mathcal{D} g_\theta - \mathbb{E}_{\mathcal{D}'} g_\theta\|$, where $g_\theta$ is a test function parameterized by $\theta$. The $\mathcal{H}$-divergence uses $g_\theta = \ell(f_\theta(x), y)$ and the $\mathcal{H}\Delta\mathcal{H}$ divergence first extends the parameter space $\Theta$ to the product space $\Theta \otimes \Theta$ and let $g_{(\theta,\theta')} = \ell(f_\theta(x), f_{\theta'}(x))$ for any $(\theta, \theta') \in \Theta \otimes \Theta$. The gradient variation uses $g_\theta = \nabla_\theta \ell(f_\theta, y)$. As we consider the local regret for non-convex problems where the goal is to find a first-order stationary point, using the gradient as the test function is a natural fit.

### 4.3 THE HEADROOM OF BATCH UPDATE

In the last section, we see that BU achieves the minimax regret, so at first sight it seems there is no room for further improvement. However, we note that this only implies that BU is optimal in the *worst-case sense*, i.e., when the future data distribution is completely uncorrelated with the previous ones. This is hardly the case in reality: the drift in the data distribution normally happens in a gradual manner, and the data distribution in the past should be informative of the

---

[1]Such overlapping mechanism is the key to defending adversaries in non-convex games and we refer to Section 2.3 in [Hazan et al., 2017] for more details.

---

**Algorithm 3** Meta Gradient Descent: a helper algorithm

**Input:** The learning rate $\eta$ for updating the parameter $\theta$.
**for** $t \in [T]$ **do**
    Deploy the prediction model $f_{\theta_t}$ with parameter $\theta_t$.
    Collect the new dataset $D_t$.
    Construct the smoothed gradient generator $\bar{m}(\cdot; t+1)$.
    Initialize $\theta_{t+1}$.
    **while** $\|\bar{m}(\theta_{t+1}; t+1)\| \geq \delta$ **do**
        $\theta_{t+1} \leftarrow \theta_{t+1} - \eta \bar{m}(\theta_{t+1}; t+1)$
    **end while**
**end for**

---

future. Hence, the natural question is: can we do better than BU in a gradually changing environment?

The discussion after Proposition 1 suggests that the only hope for improvement lies in reducing the domain generalization error $V_w(T)$. To illustrate the headroom, we start with Meta Gradient Descent (MGD), a 'helper algorithm' that extends BU and serves as an intermediate step towards the proposed FGD. Assume that we are given a sequence of gradient generators $\{m(\cdot; t)\}_{t=1}^{T}$. Then FGD uses a smoothed gradient generator given by

$$\bar{m}(\theta; t) = \frac{1}{w} \left( m(\theta; t) + \sum_{i=1}^{w-1} \nabla r_{t-i}(\theta) \right),$$

for updating, yielding Algorithm 3.

By substituting $\nabla r_{t-w}(\cdot)$ for $m(\cdot, t)$, MGD reduces to BU with $b = w$. Comparing $\bar{m}(\cdot; t)$ with $\nabla u_{w,t}$, the true gradient on the test set, we see that

$$\bar{m}(\theta; t) - \nabla u_{w,t}(\theta) = \frac{1}{w}(m(\theta; t) - \nabla r_t(\theta)), \quad (5)$$

suggesting that MGD introduces a general gradient generator $m(\theta; t)$ as a proxy for $\nabla r_t(\theta)$, similar to FGD. On the other hand, we note that the gradient generator in MGD is pre-specified, while FGD parametrizes the gradient generator $m$ with $\phi$ and optimizes it on the fly.

From this perspective, BU in Algorithm 1 in fact implicitly uses $m(\cdot, t) = \nabla r_{t-w}$ to approximate $\nabla r_t$, which explains why $V_w(T)$ depends on the difference between these two terms. While such design makes sense in the very limited case where the sequence of domains is known to have a period of $w$, it might not be a savvy choice in general. To be specific, one can construct $m$ from the observed datasets $D_{t-1}, ..., D_{t-b}$ based on some mapping parameterized by $\phi \in \Phi$. For instance, such mapping can be given by a deep neural network as described in Section 3. In this way, MGD enables a mechanism that utilizes the past domains more flexibly to predict the future gradient information $\nabla r_t$ when it can be forecasted with a more general form.

**Theorem 1.** *The $w$-local regret incurred by Algorithm 3*

*satisfies*

$$\Re_w(T) \leq 2\delta^2 + \frac{2}{w^2} Q(T; m),$$

*where $Q(T; m) := \frac{1}{T} \sum_{t=1}^{T} \sup_\theta \|\nabla r_t(\theta) - m(\theta; t)\|^2$. Furthermore, if both $\|\nabla r_t\|$ and $\|m(\cdot; t)\|$ are upper bounded by $M < \infty$ for all $\theta \in \Theta$ and $t \geq 0$, we recover the minimax regret $\Re_w(T) = O(1/w^2)$ when $\delta = O(1/w)$.*

Theorem 1 shows that we can greatly improve the regret of BU by reducing the domain generalization error $Q(T; m)$ if $m$ is properly chosen. Specifically, suppose that $\mathcal{M}$—the hypothesis class of $m$—is rich enough to model the dynamic of the data distribution, in the sense that there exists $m^* \in \mathcal{M}$ satisfying

$$Q(T; m^*) := \frac{1}{T} \sum_{t=1}^{T} \sup_\theta \|\nabla r_t(\theta) - m^*(\theta; t)\|^2 = O\left(\frac{1}{T}\right).$$

Then the domain generalization error of MGD equipped with $m^*$ tends to zero at the rate of $1/T$, in contrast to being a non-vanishing dominant term in BU. On the other hand, we can still maintain essentially the same regret bound as BU in the worst case, and thus the improvement almost comes for free.

In the following section, we show that it is indeed possible for FGD to achieve a comparable local regret bound as the one given by MGD with the optimal gradient generator $m^*$ in $\mathcal{M}$.

### 4.4 REGRET BOUND OF FGD

To simplify the analysis, we consider the case where the gradient generator at round $t$ is given by a linear model:

$$m(\theta; \phi, t) = \sum_{i=1}^{b} a_i \nabla r_{t-i}(\theta), \qquad (6)$$

where $\phi = [a_1, ..., a_b] \in S_b$ is the parameter. The hypothesis class $\mathcal{M}$ is thus $\mathcal{M} = \{\{m(\cdot; \phi, t)\}_{t=1}^{T} : \sum_{i=1}^{b} a_i \nabla r_{t-i}(\cdot), \phi \in S_b\}$. This family of FGD algorithm covers the BU algorithm, which corresponds to setting $a_b = 1$ and $a_i = 0$ otherwise. For this toy example, we use the classic exponentiated gradient descent method [Kivinen and Warmuth, 1997] to update $\phi$, which ensures that $\phi \in S_b$. The detailed algorithm is summarized in Algorithm **??** in Appendix **??**.

**Theorem 2.** *Assume that for any $t$, $\|\nabla r_t\|$ is bounded by $M < \infty$. Let $\mathcal{M}$ be the hypothesis class of $m$ given in* (6). *For any given constant $c > 0$, if we set the learning rate for updating $m$ as $\eta_\phi = c\sqrt{(\log b)/(TM^4)}$, the $w$-local regret incurred by Algorithm **??** in Appendix **??** satisfies*

$$\Re_w(T) \leq 2\delta^2 + \frac{2}{w^2}(Q(T; m^*) + O(M^2\sqrt{\log b/T})),$$

*where $Q(T; m^*) = \min_{m \in \mathcal{M}} \sum_{t=1}^{T} \sup_\theta \|\nabla r_t(\theta) - m(\theta; \phi, t)\|^2$.*

Theorem 2 suggests that FGD with optimized MFGG is able to achieve the regret of Algorithm 3 using $m^*$ with $O(1/\sqrt{T})$ excessive error. As $T$ is usually large, we can see that the excessive error is small.

## 5 RELATED WORK

**Domain Generalization.** Our problem can be viewed as an extension of the classic domain generalization problem. In short, the classic domain generalization problem that is extensively studied in vision or NLP is *one-shot* in the sense that it aims to generalize a model to one unseen target domain by training over multiple source domains. In contrast, our problem is $T$-*shot*, since we have a stream of $T$ pairs of target/source domains. The difference between one-shot and $T$-shot can be significant. In the one-shot setting, we are unable to receive feedback on how the model generalizes on the unseen domain and thus the existing algorithms are hence focus on improving the worst-case generalization by learning domain-invariant representation based on methods such as domain feature alignment [Guo et al., 2019, Li et al., 2018b], causal learning [Arjovsky et al., 2019, Wang et al., 2022b], multi-task learning [Carlucci et al., 2019], meta-learning [Balaji et al., 2018, Li et al., 2018a] and data augmentation [Ilse et al., 2021, Yan et al., 2020]. In comparison, our algorithm mainly focuses on how to use the feedback in the $T$-shot setting to learn to predict the gradient information of the future unseen domain. While adopting the techniques from the one-shot domain generalization is of interest, the design of those algorithms utilizes a lot of domain knowledge from CV or NLP, making it non-trivial to apply to recommendation systems. We thus leave it for future work.

**Continual Learning.** Continual learning is a similar scenario where the goal is to learn an accurate model given a stream of different tasks/domains. Compared with multi-task learning [Crawshaw, 2020, Sener and Koltun, 2018, Wang et al., 2021, Ye and Liu, 2021], the key challenge of continual learning is *catastrophic forgetting* [Kirkpatrick et al., 2017]: the model forgets how to solve past tasks after it is exposed to new tasks. Various of types of solutions are proposed, including rehearsal-based methods [Aljundi et al., 2019, Chaudhry et al., 2020, Lopez-Paz and Ranzato, 2017], knowledge distillation [Rebuffi et al., 2017], regularization [Buzzega et al., 2020, Kirkpatrick et al., 2017] and architecture adjustment [Rusu et al., 2016, Serra et al., 2018]. Although the learning scenario is similar, a direct application of continual learning methods to our setting might not give a desirable outcome. The reason is that the final goals of the two problems are quite different: continual learning aims to learn the current task without sacrificing the performance of the past learned tasks, while we only focus on performing well in the unobserved future task.

**Gradual Domain Adaptation**  Gradual domain adaptation (GDA) aims at adapting a model to an unlabeled target domain after being trained on a labeled source domain and a sequence of unlabeled intermediate domains. Despite being similar to the setting of temporal domain generalization, GDA is still different from the latter since there are no labels provided in the intermediate domains for GDA. A modern and common approach for GDA is gradual self-training [Dong et al., 2022, Kumar et al., 2020, Wang et al., 2022a, Zhou et al., 2022], which fits a model to the source domain and then adapts the model along the sequence of intermediate domains consecutively with self-training [Nigam and Ghani, 2000].

**Meta-Learning.**  Meta-learning, or learning-to-learn, aims to optimize the training process such that the outcome is improved. Examples of meta-learning includes learning a better initialization [Finn et al., 2017, Lee and Choi, 2018], optimizer [Andrychowicz et al., 2016, Flennerhag et al., 2019], hyper-parameter [Chen et al., 2019, Franceschi et al., 2018] and network architecture [Liu et al., 2018, Wang et al., 2022c]. The proposed FGD can be viewed as *learning a better optimizer* for the temporal domain generalization problems. Meta-learning is also widely deployed in recommendation systems. Examples include solving cold start issue [Bharadhwaj, 2019, Lee et al., 2019] through learning initialization and knowledge transferring through model fusion [Peng et al., 2021, Zhang et al., 2020].

## 6 EXPERIMENT

We demonstrate the effectiveness of the proposed FGD.

**Dataset.**  We consider two datasets CriteoTB and Avazu. CriteoTB has 13 integer feature fields and 26 categorical feature fields with around 800 million categorical tokens in total. It is the 24-day advertising data published by criteo. Training with the original CriteoTB dataset takes huge computational cost and to reduce computational overhead and increase reproducibility, we use a subsampled CriteoTB with 10% of examples are sampled for evaluation. Avazu contains 11 days of clicks/not clicks data from Avazu and all its 22 feature fields are categorical. We preprocess both datasets following Guo et al. [2017], Liu et al. [2020].

**Training Protocol.**  In real world recommendation systems, passing the examples multiple times for training might cause severe over-fitting issue [Du et al., 2021, Ye et al., 2020, Zheng et al., 2020]. Following Ye et al. [2020], Zheng et al. [2020] we perform a single pass on the training data in the sense that each training example is only visited once throughout the training. Thus, we set $\theta_t^0 = \theta_{t-b}$ during the model training at time $t$ because examples from domain $D_s$, $s \leq t - b$ has been visited for learning $\theta_{t-b}$. In Algorithm 2, the default scheme trains the recommendation models until the norm of the gradient is smaller than a threshold while in

the experiment, we use the alternative strategy in which we train the model with a fixed number of iterations such that all the examples are passed exactly once.

**Evaluation Protocol.**  As we consider an online learning environment, there is no need to split the dataset to training and testing subset. Instead, at the training time of $\theta_t$, the data at the next day $D_{t+1}$ is used to evaluate the performance of $f_{\theta_t}$ and hence the domain generalization error is considered. Such evaluation protocol matches the real recommendation systems [Ye et al., 2020]. We adopt AUC (Area Under the ROC Curve) and Logloss to measure the performance. For Criteo1TB we evaluate the performance using the last 8 or 16 days and the first 16 or 8 days are considered to be offline training for warm up start. For Avazu, the first 3 days are treated to be offline training and hence only the last 8 days are used for evaluation. The metrics are averaged over all the days that are used for evaluation. For all the experimental settings, we run all the compared approaches 3 times with different random seeds and report the averaged result.

**Models and Optimizers.**  We consider two representative architectures for recommendation models, FM [Rendle, 2010] and DeepFM [Guo et al., 2017]. Following Guo et al. [2017], Liu et al. [2020], we use Adam as our optimizer and tune the learning rate for each compared methods from $\{0.01, 0.001, 0.0001, 0.00001\}$ using the performance of the offline training and the batch size is set to be 1024. For FGD, we add the model at the training trajectory into trajectory buffer every 150/50 iterations for CriteoTB/Avazu. The meta network is trained using SGD with learning rate 0.01 and batch size 20.

**Baselines.**  For comparison, we consider the following optimization algorithms: Incremental Update (IU) [Wang et al., 2020] that updates the model incrementally only using the newly observed data $D_t$; Batch Update (BU-$b$) [Wang et al., 2020] that updates the model using the most recent $b$ domains $\{D_t, ..., D_{t+1-b}\}$; Stream-centered Probabilistic Matrix Factorization (SPMF-$b$) [Wang et al., 2018] in which a reservoir of historical examples are maintained to mix with the new data for current model updating. SPMF-$b$ denotes the setting that the example buffers has the same size as the number of examples in $b$ days; Adaptive Sequential Model Generation (ASMG-$b$) [Peng et al., 2021] that generates a better serving model from a sequence of $b$ most recent historical serving models via a meta generator; Future Gradient Descent (FGD-$b$) is our approach with the recent $b$ domains used for training the recommendation models.

**Result.**  Table 1 and 2 summarized the results for CriteoTB and Avazu, respectively. The proposed FGD out-performs the baselines in most cases. We also observe that increasing $b$ improves the performance for most algorithms as more information can be utilized. The performance boost of FGD when increasing $b$ is more significant than other approaches.

| Method | FM | | | | DeepFM | | | |
|---|---|---|---|---|---|---|---|---|
| | Auc-8 ↑ | Logloss-8 ↓ | Auc-16 ↑ | Logloss-16 ↓ | Auc-8 ↑ | Logloss-8 ↓ | Auc-16 ↑ | Logloss-16 ↓ |
| IU | $60.35 \pm 0.54$ | $16.74 \pm 0.25$ | $60.56 \pm 0.61$ | $16.78 \pm 0.16$ | $60.48 \pm 0.47$ | $15.63 \pm 0.19$ | $60.62 \pm 0.60$ | $15.69 \pm 0.12$ |
| BU-2 | $62.69 \pm 0.50$ | $16.04 \pm 0.19$ | $62.31 \pm 0.73$ | $16.15 \pm 0.20$ | $62.65 \pm 0.37$ | $15.21 \pm 0.15$ | $62.40 \pm 0.48$ | $15.22 \pm 0.13$ |
| SPMF-2 | $61.56 \pm 0.43$ | $18.30 \pm 0.31$ | $61.41 \pm 0.75$ | $18.48 \pm 0.20$ | $61.12 \pm 0.57$ | $15.74 \pm 0.21$ | $60.64 \pm 0.90$ | $15.65 \pm 0.13$ |
| ASMG-2 | $63.82 \pm 0.42$ | $16.51 \pm 0.28$ | $63.80 \pm 0.49$ | $16.54 \pm 0.19$ | $63.95 \pm 0.42$ | $15.00 \pm 0.19$ | $63.85 \pm 0.54$ | $\mathbf{14.96 \pm 0.13}$ |
| Meta-2 | $\mathbf{65.23 \pm 0.46}$* | $\mathbf{15.81 \pm 0.27}$* | $\mathbf{64.84 \pm 0.61}$* | $\mathbf{15.89 \pm 0.20}$* | $\mathbf{65.04 \pm 0.42}$* | $\mathbf{14.93 \pm 0.16}$* | $\mathbf{64.60 \pm 0.57}$* | $14.96 \pm 0.13$ |
| BU-3 | $63.55 \pm 0.46$ | $15.28 \pm 0.15$ | $63.40 \pm 0.64$ | $15.30 \pm 0.13$ | $63.65 \pm 0.40$ | $14.93 \pm 0.14$ | $63.41 \pm 0.51$ | $14.90 \pm 0.11$ |
| SPMF-3 | $60.73 \pm 0.55$ | $18.18 \pm 0.35$ | $61.00 \pm 0.87$ | $18.32 \pm 0.23$ | $61.83 \pm 0.54$ | $14.99 \pm 0.16$ | $61.32 \pm 0.62$ | $14.74 \pm 0.12$ |
| ASMG-3 | $63.21 \pm 0.49$ | $18.51 \pm 0.41$ | $63.35 \pm 0.69$ | $19.61 \pm 0.27$ | $65.02 \pm 0.41$ | $14.82 \pm 0.17$ | $64.77 \pm 0.53$ | $14.80 \pm 0.11$ |
| Meta-3 | $\mathbf{67.20 \pm 0.25}$* | $\mathbf{15.09 \pm 0.18}$* | $\mathbf{67.05 \pm 0.38}$* | $\mathbf{15.10 \pm 0.14}$* | $\mathbf{66.92 \pm 0.26}$* | $\mathbf{14.65 \pm 0.15}$* | $\mathbf{66.78 \pm 0.37}$* | $\mathbf{14.62 \pm 0.11}$ |
| BU-5 | $66.19 \pm 0.24$ | $14.76 \pm 0.18$ | $66.24 \pm 0.30$ | $14.71 \pm 0.13$ | $66.15 \pm 0.23$ | $14.54 \pm 0.15$ | $66.23 \pm 0.29$ | $14.49 \pm 0.11$ |
| SPMF-5 | $61.96 \pm 0.44$ | $14.69 \pm 0.13$ | $62.21 \pm 0.53$ | $14.74 \pm 0.10$ | $63.79 \pm 0.41$ | $14.83 \pm 0.18$ | $62.79 \pm 0.48$ | $14.53 \pm 0.13$ |
| ASMG-5 | $65.82 \pm 0.32$ | $14.79 \pm 0.14$ | $65.99 \pm 0.40$ | $14.79 \pm 0.11$ | $66.49 \pm 0.26$ | $14.50 \pm 0.14$ | $66.47 \pm 0.35$ | $14.50 \pm 0.10$ |
| Meta-5 | $\mathbf{69.00 \pm 0.21}$* | $\mathbf{14.62 \pm 0.13}$ | $\mathbf{69.37 \pm 0.19}$* | $\mathbf{14.61 \pm 0.11}$ | $\mathbf{68.85 \pm 0.33}$* | $\mathbf{14.39 \pm 0.23}$ | $\mathbf{69.15 \pm 0.28}$* | $\mathbf{14.38 \pm 0.22}$ |

Table 1: Summarized result for CriteoTB. AUC/Logloss-x denotes the resulted based on the last x days examples. The averaged performance over three random seeds with its standard deviation are reported. We mainly compare the algorithm when the same $b$ is used and the best approach as bolded. The * denotes that the best result are statistically significant compared with the second best with p value less than 0.95 using matched-pair t-test.

| Method | FM | | DeepFM | |
|---|---|---|---|---|
| | Auc ↑ | Logloss ↓ | Auc ↑ | Logloss ↓ |
| IU | $73.82 \pm 0.18$ | $39.92 \pm 0.86$ | $73.99 \pm 0.22$ | $39.80 \pm 0.81$ |
| BU-2 | $74.16 \pm 0.25$ | $39.71 \pm 0.88$ | $74.31 \pm 0.21$ | $39.59 \pm 0.86$ |
| SPMF-2 | $69.31 \pm 0.31$ | $45.51 \pm 0.99$ | $71.11 \pm 0.53$ | $42.09 \pm 0.59$ |
| ASMG-2 | $\mathbf{74.22 \pm 0.20}$ | $\mathbf{39.66 \pm 0.89}$ | $\mathbf{74.34 \pm 0.19}$ | $39.58 \pm 0.85$ |
| Meta-2 | $\mathbf{74.22 \pm 0.28}$ | $39.77 \pm 0.90$ | $\mathbf{74.34 \pm 0.21}$ | $\mathbf{39.54 \pm 0.87}$ |
| BU-3 | $74.17 \pm 0.31$ | $\mathbf{39.68 \pm 0.89}$ | $74.50 \pm 0.30$ | $39.48 \pm 0.90$ |
| SPMF-3 | $68.95 \pm 0.56$ | $47.17 \pm 1.27$ | $71.93 \pm 0.24$ | $41.83 \pm 0.64$ |
| ASMG-3 | $73.64 \pm 0.08$ | $39.93 \pm 0.83$ | $73.95 \pm 0.17$ | $39.82 \pm 0.83$ |
| Meta-3 | $\mathbf{74.20 \pm 0.27}$* | $\mathbf{39.68 \pm 0.89}$ | $\mathbf{74.55 \pm 0.28}$* | $\mathbf{39.45 \pm 0.90}$ |

Table 2: Summarized result for Avazu. The setting of the table is the same as that of Table 1.

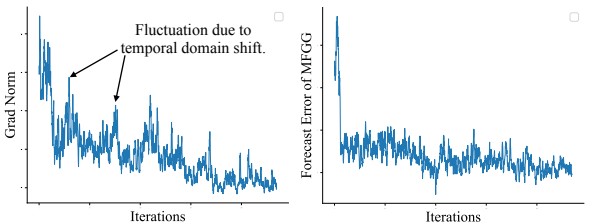

Figure 3: Left: evolution of $\|\nabla r_t(\theta_{t,i})\|^2$. Right: the normalized forecast error of MFGG in different time and iterations.

Compared with CriteoTB, FGD is less significantly better in Avazu dataset. We think the reason might be that the domains of different days in Avazu are less different compared with that in CriteoTB.

**Temporal Domain Shift and Forecast Error of MFGG.** To visualize the effect of the temporal domain shift, we plot the gradient norm during the whole training process. We consider FGD-3 in CriteoTB with DeepFM as the recommendation models. In this examples, at each time $t$, the recommendation model is trained with $R = 20K$ iterations. At time $t-1$, denote $\theta_{t,i}$ as the parameter at the $i$-th iteration of the training (note that after the training $\theta_t$ is used to predict examples in $D_t$). We visualize the evolution of the gradient norm of the future domain $g_{t,i} = \|\nabla r_t(\theta_{t,i})\|^2$ in a chrono-

logical order (i.e., $..., g_{t,1}, ..., g_{t,R}, g_{t+1,1}, ..., g_{t,R}, ...$) in the left subfigure of Fig 3. Overall, $g$ is decreasing suggesting the improving performance but significant fluctuation of $g$ is also observed: when we shift from $t$ to $t+1$, $g$ will suddenly increase demonstrating a considerable deviation between the adjacent domains. We also visualize the (normalized) forecast error $e_{i,t}$ of MFGG in the right subfigure of Fig 3

$$e_{t,i} = \frac{\|m(\theta_{t+1,i}; \phi_t, t) - \nabla r_{t+1}(\theta_{t+1,i})\|^2}{\|\nabla r_{t+1}(\theta_{t+1,i})\|^2}.$$

Here, we normalize the error by the gradient norm $\|\nabla r_{t+1}(\theta_{t+1,i})\|^2$ to rule out the effect of the decrease of gradient norm. We observe a decrease of the forecast error demonstrating that the gradient of future domain can be predicted using the past domains. Besides, the error remains stationary which provides evidence that the modeling the MFGG as a functional time-series model is reasonable.

**Optimizing MFGG with Random Model.** When optimizing MFGG, the loss is calculated based on a model $f_\theta$ sampled from its training trajectory so that we make MFGG focus on giving good prediction on the gradient of $f_\theta$ that has reasonable performance. To show the importance of such design, we also run FGD in which MFGG is optimized using $f_\theta$ with $\theta$ randomly initialized. We consider the setting of FGD-3 in CriteoTB and use both FM and DeepFM as recommendation model and summarize the result in Table 3. It can be shown that train the MFGG with random recommendation model degenrates the performance.

**Computation Overhead.** We compare the wall clock training time of BU and FGD. We consider the DeepFM model in CriteoTB and report the averaged training time with different $b$ at each time $t$ in Table 4. It can be shown that the proposed FGD introduces only about 15% overhead.

| Buffer | Method | Auc-8 ↑ | Logloss-8 ↓ | Auc-16 ↑ | Logloss-16 ↓ |
|--------|--------|---------|-------------|----------|--------------|
| FM | Rand | $67.08 \pm 0.28$ | $15.17 \pm 0.21$ | $67.08 \pm 0.41$ | $15.28 \pm 0.16$ |
| | Traj | $\mathbf{67.20 \pm 0.25}$ | $\mathbf{15.09 \pm 0.18}$ | $67.05 \pm 0.38$ | $\mathbf{15.10 \pm 0.14}$ |
| DeepFM | Rand | $66.83 \pm 0.27$ | $14.68 \pm 0.16$ | $66.68 \pm 0.41$ | $14.66 \pm 0.12$ |
| | Traj | $\mathbf{66.92 \pm 0.26}$ | $14.65 \pm 0.15$ | $\mathbf{66.78 \pm 0.37}$ | $\mathbf{14.62 \pm 0.11}$ |

Table 3: Comparing the performance when MFGG is trained with model sampled from optimization trajectory (Traj) and randomly initialized model (Rand). The setting of the table is the same as that of Table 1.

| Time/min | BU-2 | Meta-3 | BU-3 | Meta-3 | BU-3 | Meta-23 |
|----------|------|--------|------|--------|------|---------|
| | 20.4 | 24.3 | 29.7 | 33.8 | 47.6 | 52.2 |

Table 4: Comparing the wall clock training time of BU and FGD at each round ($t$).

# 7 CONCLUSION

In this paper, we propose future gradient descent (FGD) that forecasts the gradient information of the future domain for training to address the issue of temporal domain shift in online recommendation systems. We show that FGD gives smaller temporal domain generalization in theory compared with a widely adopted algorithm, Batch Update. Empirical evidence is provided to show that FGD outperforms various representatives algorithms.

**Acknowledgements**

This work is supported by grant from Meta Inc.

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
