# OpenReview forum: "Future Gradient Descent for Adapting the Temporal Shifting Data Distribution in Online Recommendation System"
_auai.org/UAI/2022/Conference — UAI 2022 Poster_

### Official Review · Reviewer_jpj1 · 2022-04-11

**Q2(1) Originality/Novelty:** 3
**Q2(2) Significance/Impact:** 2
**Q2(3) Correctness/Technical Quality:** 3
**Q2(6) Clarity Of Writing:** 2
**Q6 Overall Score:** 5
**Q8 Confidence In Your Score:** 3

**Q1 Summary And Contributions:**

The temporal domain shift is an important problem in machine learning. The authors propose the usage of a meta future gradient generator to predict the gradient of future data in order to address this problem in the context of recommender systems. It has better performance compared to baseline methods such as Batch Update on CriteoTB and Avazu datasets. The authors also provide the corresponding theoretical analysis.

**Q2 Assessment Of The Paper:**

More detailed information regarding each of these aspects is given below:

**Q2(4) Quality Of Experiments (Optional):**

2: Fair: The experimental evaluation is weak: important baselines are missing, or the results do not adequately support the main claims.

**Q2(5) Reproducibility:**

2: Fair: Key resources (e.g., proofs, code, data) are unavailable but key details (e.g., proof sketches, experimental setup) are sufficiently well-described for an expert to confidently reproduce the main results.

**Q3 Main Strengths:**

The main contribution of this article is to propose a new solution to the problem of temporal domain shift in the context of recommendation system. The domain distribution changes over time and may also differ in different splits of the data. As we have no associated prior knowledge of such shifts, the model undergoes generalization errors in this case. Tactfully, authors exploit meta learning to model this shift and present the idea of using generators to produce gradients. The motivation is sensible. Their approach is also validated on two real datasets, and a corresponding theoretical analysis is given.

**Q4 Main Weakness:**

See the detailed comments below.

**Q5 Detailed Comments To The Authors:**

Interesting idea proposed by the author, but there are some concerns which I hope the author will elaborate on.
- Central to this article is the idea of using meta learning methods to predict characteristics of future data that can be used in a direct or secondary way to help update the model. The authors predict the gradient directly based on meta learning. I don't know if it is possible to predict alternative types of knowledge, such as label shift. Also is it possible to anticipate hyper-parameters other than gradient, such as learning rate?
- Simulation of the generation of gradients for future data based on meta learning is predicated on the assumption that the domain bias is small. However, in practice, it is possible that at some moment in time, the domain shift may be substantial, will the predicted gradient be unbiased?
- Authors only compare to baseline methods with final experimental results. I kind of wish the authors would perform some ablation studies. For illustration, we can indeed acquire the true gradients, would the author show the difference between the predicted gradients and the true gradients?
- The authors performed experiments on two datasets. I expect the authors to compete the methods on more datasets.
- There is a lack of detail in the experiments, which leads to reproduction difficulties.

In addition, there are some typos and poor presentation throughout the submission, and the author needs to carefully check them.
typos:
- "interested in make" should be "in making"
- "by apply" should be "by applying"
- By and large, the motivation, solutions and theoretical analysis of this post are commendable. But there are still some issues that need to - be addressed. The authors need to settle them to make this submission more thorough.

**Q7 Justification For Your Score:**

The paper is well-motivated, but the experiments can be improved (more datasets and more implementation details).


**Q9 Complying With Reviewing Instructions:**

1: Yes.

---

### Official Review · Reviewer_YqQx · 2022-04-12

**Q2(1) Originality/Novelty:** 2
**Q2(2) Significance/Impact:** 2
**Q2(3) Correctness/Technical Quality:** 3
**Q2(6) Clarity Of Writing:** 4
**Q6 Overall Score:** 7
**Q8 Confidence In Your Score:** 3

**Q1 Summary And Contributions:**

Solving the temporal domain generalization problem by forecasting the future training information.

**Q2 Assessment Of The Paper:**

More detailed information regarding each of these aspects is given below:

**Q2(4) Quality Of Experiments (Optional):**

3: Good: The experimental evaluation is adequate, and the results convincingly support the main claims.

**Q2(5) Reproducibility:**

3: Good: Key resources (e.g., proofs, code, data) are available and key details (e.g., proofs, experimental setup) are sufficiently well-described for competent researchers to confidently reproduce the main results.

**Q3 Main Strengths:**

Solid theoretical basis for the proposed approach.

Very clear presentation and development of ideas. Each step in the process of formulating their solution is well motivated. An enjoyable read.

Good experiments, good explanation of results.

Figures 1 and 2 are brilliant visual summaries of the different approaches.



**Q4 Main Weakness:**

Not a game-changing technology. Incremental enhancements to solving a very specific learning problem. May not be of interest to the broad community.

**Q5 Detailed Comments To The Authors:**

Is a hybrid approach possible which combines the direct and indirect approaches you mention at the outset? Or is your approach in effect a hybrid approach?

**Q7 Justification For Your Score:**

I've read the paper carefully and could not find errors, but I'm not familiar with the state of the art of approaches to solving the problem, so I might have missed some technical detail.

**Q9 Complying With Reviewing Instructions:**

1: Yes.

---

### Official Review · Reviewer_n8Mh · 2022-04-16

**Q2(1) Originality/Novelty:** 2
**Q2(2) Significance/Impact:** 2
**Q2(3) Correctness/Technical Quality:** 3
**Q2(6) Clarity Of Writing:** 3
**Q6 Overall Score:** 5
**Q8 Confidence In Your Score:** 1

**Q1 Summary And Contributions:**

The paper presents an approach for dealing with temporal domain shift in the context of online recommendation by forecasting the gradient of the future examples.

**Q2 Assessment Of The Paper:**

More detailed information regarding each of these aspects is given below:

**Q2(4) Quality Of Experiments (Optional):**

2: Fair: The experimental evaluation is weak: important baselines are missing, or the results do not adequately support the main claims.

**Q2(5) Reproducibility:**

3: Good: Key resources (e.g., proofs, code, data) are available and key details (e.g., proofs, experimental setup) are sufficiently well-described for competent researchers to confidently reproduce the main results.

**Q3 Main Strengths:**

The topic is relevant and the proposal is well described. Experiments seems to suggest that the approach is sound.

**Q4 Main Weakness:**

The experimental evaluation is done over data of short periods of time with sub-samples of datasets. The impact on real recommender systems need to be further explained.

**Q5 Detailed Comments To The Authors:**

The paper presents an approach for dealing with temporal domain shifts by forecasting the future gradient descent. A paper can be greatly improved with a discussion of the impact of this approach in real-word recommenders and domains in which this can be potentially applied. Also, the use to forecast more gradual concept-drift, as the ones occurring in user profiling, can be analyzed. The novelty of the approach need to be highlighted in relation with state-of-the-art research. Experimental results have also room for improvement and requires a more detailed explanation. The selection of a small sample (10%) of CriteoTB, for example, it is not justified enough. The evaluation is done over short periods of time (a few days), the application of the approach over longer periods of time needs to be also discussed.

**Q7 Justification For Your Score:**

The paper can improve the experimental validation and highlith the contribution.

**Q9 Complying With Reviewing Instructions:**

1: Yes.

---

### Official Review · Reviewer_XxGG · 2022-04-19

**Q2(1) Originality/Novelty:** 3
**Q2(2) Significance/Impact:** 2
**Q2(3) Correctness/Technical Quality:** 3
**Q2(6) Clarity Of Writing:** 3
**Q6 Overall Score:** 6
**Q8 Confidence In Your Score:** 3

**Q1 Summary And Contributions:**

To take the temporal domain shift challenge for online recommendation, this paper proposed propose to learn a meta future gradient generator algorithm. The theoretical analysis showed that smaller temporal domain generalization error measured by a gradient variation term in a local regret can be achieved. The empirical experiments demonstrated the advantage of the proposed algorithm.

**Q2 Assessment Of The Paper:**

More detailed information regarding each of these aspects is given below:

**Q2(4) Quality Of Experiments (Optional):**

3: Good: The experimental evaluation is adequate, and the results convincingly support the main claims.

**Q2(5) Reproducibility:**

2: Fair: Key resources (e.g., proofs, code, data) are unavailable but key details (e.g., proof sketches, experimental setup) are sufficiently well-described for an expert to confidently reproduce the main results.

**Q3 Main Strengths:**

1. This paper deals with an interesting problem. The proposed idea is somewhat novel.
2. The organizations are clear.

**Q4 Main Weakness:**

Minor things:
1. The presentation can be further improved. 1) There exist some grammar errors in the presentation.  2) Sometimes, the pointers of the cited papers are not working well. The readers feel difficulty to find the correct references when clicking the pointers.

**Q5 Detailed Comments To The Authors:**

1.The paper proposed a novel perspective by framing the problem of learning under shifting domains as a temporal domain generalization problem. It works with the gradient by introducing a meta future gradient generator. It is a direct solution towards the temporal domain generalization problem based on forecasting the future information for training, train a meta future gradient generator using FGD update.

2.The overall presentation is clear and well-organized. For example, the paper provides a specific statement in the background about the Temporal Domain Generalization and BU, a widely used updating pipeline. The paper also provides a good illustration of three different updating methods in Figure 2, presenting the difference among these methods, which helps the reader understand their main ideas quickly.

 3. A theoretical generalization error analysis is provided. The readers can also get some insights from their analysis.

4. The experiments seem to be solid.  This paper also provides a good statement about the experiment, including the training and evaluating procedure and different baselines and settings. The experiments show the advantage of the proposed method.

5. Minor.
Grammar mistakes.

"We are more
interested in make MFGG accurate at a small subset of the
model parameter space..." --> interested in making

Pointers of the cited paper.
In your experiments baseline descriptions.
The pointers of your baselines are not working well.

**Q7 Justification For Your Score:**

1. This paper deals with an interesting problem. The proposed idea is somewhat novel.
2. The organizations are clear.

**Q9 Complying With Reviewing Instructions:**

1: Yes.

---

### Decision · Program_Chairs · 2022-05-15

**Decision:**

Accept (Poster)

**Comment:**

Meta Review: The reviewers are all positive about this paper. The proposed idea is somewhat novel and the paper is well presented.  One suggestion is to evaluate the proposed approach on more sophisticated datasets.